# A New Era in Ocular Therapeutics: Advanced Drug Delivery Systems for Uveitis and Neuro-Ophthalmologic Conditions

**DOI:** 10.3390/pharmaceutics15071952

**Published:** 2023-07-14

**Authors:** Kevin Y. Wu, Kenneth Tan, Dania Akbar, Mazen Y. Choulakian, Simon D. Tran

**Affiliations:** 1Department of Surgery, Division of Ophthalmology, University of Sherbrooke, Sherbrooke, QC J1G 2E8, Canada; yang.wu@usherbrooke.ca (K.Y.W.);; 2Faculty of Medicine and Health Sciences, McGill University, Montreal, QC H3T 1J4, Canada; 3Department of Human Biology, University of Toronto, Toronto, ON M5S 1A1, Canada; 4Faculty of Dental Medicine and Oral Health Sciences, McGill University, Montreal, QC H3A 1G1, Canada

**Keywords:** uveitis, neuro-ophthalmology, drug-delivery systems (DDS), polymeric nano-based DDS, ocular barriers, biodegradability, bioavailability, biopolymers, biodegradable implant, targeted drug delivery

## Abstract

The eye’s intricate anatomical barriers pose significant challenges to the penetration, residence time, and bioavailability of topically applied medications, particularly in managing uveitis and neuro-ophthalmologic conditions. Addressing this issue, polymeric nano-based drug delivery systems (DDS) have surfaced as a promising solution. These systems enhance drug bioavailability in hard-to-reach target tissues, extend residence time within ocular tissues, and utilize biodegradable and nanosized polymers to reduce undesirable side effects. Thus, they have stimulated substantial interest in crafting innovative treatments for uveitis and neuro-ophthalmologic diseases. This review provides a comprehensive exploration of polymeric nano-based DDS used for managing these conditions. We discuss the present therapeutic hurdles posed by these diseases and explore the potential role of various biopolymers in broadening our treatment repertoire. Our study incorporates a detailed literature review of preclinical and clinical studies from 2017 to 2023. Owing to advancements in polymer science, ocular DDS has made rapid strides, showing tremendous potential to revolutionize the treatment of patients with uveitis and neuro-ophthalmologic disorders.

## 1. Introduction

Managing uveitis and neuro-ophthalmologic conditions presents a significant challenge due to the eye’s complex anatomy, which restricts effective medication delivery. Conventional therapies such as topical eye drops and intravitreal injections face limitations due to their poor bioavailability, short residence time, and the need for frequent dosing.

To circumvent these constraints, the development of biodegradable nano-based drug delivery systems (DDS) has gained prominence. These systems promise extended residence time in ocular tissues, improved penetration through ocular barriers, and are composed of nanosized, biodegradable polymers, thereby diminishing the risk of toxicity and adverse reactions.

In this review, we offer a thorough overview of recent advancements in biodegradable nano-based DDS for the treatment of uveitis and neuro-ophthalmologic conditions. We scrutinize the current therapeutic challenges and investigate various types of biodegradable nanocarriers, highlighting their potential to enhance treatment strategies. This exploration includes an extensive literature review of preclinical and clinical studies from 2017 to 2023, showcasing the rapid evolution of nano-based DDS.

In light of advancing biodegradable materials and a deeper comprehension of ocular pharmacology, nano-based DDS exhibits significant promise in surmounting the hurdles faced in managing uveitis and neuro-ophthalmologic disorders.

## 2. Anatomical Barriers

Administering ophthalmic medications to ocular tissues involves several methods, including topical, subconjunctival, suprachoroidal, intracameral, intravitreal, retrobulbar, sub-tenon, posterior juxta-scleral, subretinal, and systemic delivery (Figure 1). The most commonly used routes of administration are topical, systemic, periocular, and intraocular methods for most of the ocular pathologies encountered in clinical settings.

Topical application is the most straightforward approach, encompassing various preparations such as solutions, suspensions, ointments, gels, or emulsions. However, only a small fraction—around 5%—of the applied dose manages to penetrate the eye’s internal structures. The ocular barriers, including the tear film, cornea, vitreous, blood-aqueous barrier, and blood-retina barrier, restrict the uptake of solutes and fluids into the eye’s anterior and posterior parts (Figure 2). These barriers serve as protective shields against potentially harmful external molecules but simultaneously reduce ocular drug bioavailability. The blood-retina barrier primarily limits drug absorption from the systemic circulation to the posterior sections of the eye, while the other barriers predominantly counteract the absorption of externally applied drugs into the anterior and posterior segments of the eye [1].

A significant challenge in ocular drug delivery is the presence of the various anatomical barriers discussed below. Topical or systemic applications, while the simplest, present a significant challenge of successful barrier penetration with sufficient drug concentration to have a therapeutic effect as well as rapid clearance. This is particularly the case for delivery to the posterior eye segment. Alternate routes include injections into various ocular segments, which face challenges of patient compliance, frequent injections, and injection-related adverse events [2].

### 2.1. Tear Film

The tear film is composed of a complex three-layered structure: a lipid layer, an aqueous layer, and a mucin layer, all of which lie on the hydrophobic epithelium surface. Notably, the boundary between the mucous and aqueous layers is microscopically indistinct.

Topical eyedrop delivery faces a significant challenge due to the continuous removal of the drug from the eye’s surface by lacrimal fluid secretion. This clearance mechanism, in conjunction with reflex blinking, leads to substantial tear turnover. With a lacrimal fluid turnover rate of approximately 1 µL/min, a significant dose of the drug can be rapidly expelled [3].

### 2.2. Nasolacrimal Drainage System

Nearly all (95%) of a drug applied to the eye is disposed of through the nasolacrimal duct, leading from the eye to the nasal cavity, which includes the lacrimal sac, canaliculi, and nasolacrimal ducts. This process can lead to unintentional systemic drug absorption due to the vascular nature of the duct and sac, causing potential side effects and reduced drug availability to the targeted eye tissue. Factors like the topical drug’s volume, reflex blinking, and age affect this unwanted absorption. Thus, it is crucial that drug delivery design prioritize retention on the eye surface, ensuring efficient medication delivery to intended ocular areas while minimizing drainage [3].

### 2.3. Cornea

As a protective five-layer structure, the cornea limits foreign substances’ infiltration and preserves intraocular tissues. This structure comprises the epithelium, Bowman’s membrane, stroma, Descemet’s membrane, and endothelium. The corneal epithelium includes surface tight junctions and gap junctions, while the stroma and Descemet’s membrane secure the inner endothelial cells, featuring macula adherents for substance transfer [3].

In addition to its protective role, the cornea acts as a semi-permeable tissue, passively enabling material movement across its cells. It restricts the diffusion of large and hydrophilic molecules through its surface tight junctions, known as zonulae occludens, permitting only smaller molecules to pass through the 2.0 nm average diameter pores. The collagen-rich stroma is largely hydrophilic and, therefore, hinders the traversal of lipophilic molecules. Furthermore, at physiological pH, the pores’ negative charge forms an additional barrier for charged molecules due to ionic interaction [1].

Various factors like lipophilicity, molecular weight, charge, and drug ionization degree can impact transcorneal transport. Occasionally, drugs can diffuse into the aqueous humor post-transcorneal transport but fail to reach the eye’s posterior segments at therapeutic concentrations due to hindered diffusion across the vitreous humor [1].

### 2.4. Vitreous

Conversely, intravitreal drug administration offers a more direct route to the vitreous and retina. Nonetheless, the diffusion of larger and positively charged drugs may be hindered as they attempt to cross the retinal pigment epithelium (RPE) barrier to reach the choroid [1].

### 2.5. Aqueous Humor

Drugs are cleared from the aqueous humor via two methods: the conventional trabecular meshwork outflow pathway, involving the chamber angle and Schlemm’s canal, and the uveoscleral outflow pathway, via the anterior uvea’s venous blood flow. The first operates on convective flow and is drug characteristic-independent, while the second depends on the drug’s lipophilicity for crossing the vessel endothelium for elimination [4].

### 2.6. Blood-Ocular Barrier (BOB)

The blood-ocular barrier (BOB) comprises two distinct components: the blood-aqueous barrier (BAB) and the blood-retinal barrier (BRB). Together, they significantly obstruct systemic drug delivery in the anterior and posterior chambers of the eye [3].

#### 2.6.1. Blood-Aqueous Barrier (BAB)

Associated with the anterior chamber, the blood-aqueous barrier (BAB) is composed of endothelial cells, the iris, ciliary muscle, as well as pigmented and nonpigmented epithelium cells. Its tight junctions limit the entry of drug molecules [3].

#### 2.6.2. Blood-Retinal Barrier (BRB)

The blood-retinal barrier (BRB), another deterrent to drug entry from the bloodstream into the posterior chamber, comprises retinal capillary endothelium and retinal pigment epithelium cells (RPEs), which function as the inner and outer blood-retinal barriers, respectively. While it is easier to ascertain drug permeability across RPEs, quantifying permeability through retinal capillaries proves more challenging. Moreover, the particle size of the drug plays a vital role in its permeation through retinal capillaries [3].

In summary, the various anatomical barriers, ranging from the tear film to the BRB, present unique challenges to efficient ocular drug delivery. Overcoming the anatomical barriers to ocular drug delivery remains a significant challenge. However, with a comprehensive understanding of these barriers and their specific characteristics, biodegradable nano-based drug delivery systems (DDS) can be tailored more effectively to navigate these challenges, enhancing both drug bioavailability and therapeutic effectiveness.

## 3. Overview of the Biodegradable Nano-Based Drug Delivery System (DDS)

### 3.1. Enhancing Drug Delivery with Biodegradable Nanocarriers

Biodegradable nanocarriers enhance drug delivery by improving bioavailability and reducing frequent dosages, increasing patient compliance. Polymers, like cellulose derivatives, extend drug retention time, and mucoadhesive polymers limit lacrimal clearance. The use of targeting moieties directs nanocarriers to specific ocular sites, and disease-responsive designs prevent undesired drug release. These systems also enhance drug stability by modulating interactions with tear proteins and adjusting to varying eye pH levels [5].

### 3.2. Ideal Properties of Nanocarriers

Nanocarriers for drug delivery should have the ideal characteristics highlighted in Figure 3. The use of multiple polymers to form nano-assemblies, especially copolymers, provides a flexible platform for designing effective drug delivery systems with optimal properties like charge, solubility, and aggregation, and diverse shapes such as spheres, rods, or cylinders [5].

### 3.3. Exploring Various Biodegradable Polymers and Their Advantages in Ocular Drug Delivery

Ensuring the biodegradability and morphological appropriateness of polymers in nanocarriers is essential for safe and efficient drug delivery, as these characteristics allow for their natural breakdown into harmless metabolites [6]. The application of biocompatible and biodegradable polymers in ocular drug delivery systems has gained significant traction in recent years. The surface properties of these polymers, including size and charge, can influence their binding affinity and vary across different regions of the eye. Table 1 summarizes some of the most commonly used biopolymers in ocular drug delivery and highlights their unique advantages.

Hyaluronic acid, due to its negative charge and water retention capabilities, excels in enhancing the mechanical strength and drug release of hydrogels and liposome coatings [6]. Cellulose nanocrystals offer improved viscosity and varied structures for ocular drug delivery, with derivatives like carboxymethylcellulose and hydroxypropyl methylcellulose contributing to dry eye treatments and mucoadhesive enhancements, respectively [7,8,9,10]. Chitosan, with its distinctive mucoadhesive properties and permeability enhancement, can be chemically modified for additional antibacterial activity [11,12,13]. Alginate, characterized by its reversible gelation properties and functional groups, is instrumental in efficient drug encapsulation in copolymeric nanoparticles and hydrogels [14,15]. PLGA stands out for its entrapment efficiency and trans-ocular permeation, courtesy of its tunable size and surface potential [16,17]. Poloxamer 407, an FDA-approved biodegradable surfactant, is versatile in various ocular formulations [14]. Finally, cyclodextrins’ unique chemistry significantly boosts the bioavailability of numerous molecules, making them a preferred choice for ocular drug delivery [18].

### 3.4. Categories of Nano-Based DDS: Features and Improvements

Nanocarriers are tailored to deliver medicine effectively to specific eye regions by utilizing the interplay of different biodegradable polymers. This customization considers the drug’s properties and the microenvironment of the target ocular tissue. The key features are briefly summarized in Table 2 [19,20,21,22,23,24,25,26,27,28,29,30,31,32,33,34,35,36,37,38,39].

Hydrogels are garnering interest in ocular drug delivery due to their customizable pH-responsive release and target specificity, reducing the need for frequent injections [19,20]. Dendrimers, with their unique structure, efficiently encapsulate drugs and allow tunable release rates. They can be used to create other systems like dendrimer hydrogels, nanogels, and liposomes [21,22]. Liposomes, known for their vesicle structure, exhibit improved pre-corneal and conjunctival penetration. Coatings like PAMAM further enhance their permeability and bioactivity [22,23,24].

Nanomicelles, surfactant assemblies, are adept at enclosing hydrophobic compounds, facilitating corneal penetration. They are enhanced by mucin-targeting moieties and stability-enhancing cross-linking techniques [25,26,27,28,29,30]. Dispersed nanoparticles, with their self-assembling nature, offer an effective platform for drug delivery. Their size and ability to accumulate selectively in tissue make them highly suitable for ocular applications [31,32,33,34].

Nanosuspensions and nanoemulsions, used for delivering poorly soluble and permeable drugs, have shown effectiveness. Nanoemulsions boost drug solubility, while nanosuspensions, stabilized by various polymers, are ideal for high-molecular-weight drugs [35,36,37,38]. Finally, microneedles, a newer technique, offer improved accuracy, self-administration, and fewer complications from injections [39].

## 4. Biodegradable Nano-Based DDS for Uveitis

In general, non-infectious anterior uveitis is treated by applying topical glucocorticoid steroids hourly, which are then gradually tapered over several weeks once the anterior chamber inflammation is resolved. While this treatment approach is successful for the majority of patients, a subgroup does not respond favorably due to factors such as increased intraocular pressure (i.e., steroid responders), flare-ups during the tapering period, non-compliance with the treatment (such as abruptly discontinuing the steroid treatment), and ocular irritation from the frequent application of topical drugs [7].

The treatment of intermediate, posterior, and panuveitis tends to be more complex. These typically necessitate intravitreal injections because the eye’s anatomical barriers restrict the topical steroids’ bioavailability to the eye’s posterior segment. However, intravitreal injections of triamcinolone are not risk-free; they can lead to rare but severe complications like endophthalmitis and retinal detachment, as well as the inevitable adverse effects such as cataract formation and ocular hypertension if performed routinely. Additionally, they necessitate frequent follow-up office visits for re-injection, relying heavily on patients’ compliance. Intravitreal implants offering sustained corticosteroid release have been explored, but limitations such as no substantial long-term benefits, potential anterior migration (which leads to corneal decompensation), and exceedingly high rates of cataracts and ocular hypertension exist. These often necessitate cataract surgery and intraocular pressure-lowering agents [7].

In cases of bilateral disease, oral corticosteroids are an option. However, these are often associated with numerous systemic adverse effects, including fatigue, muscle weakness, weight gain, insomnia, and an increased risk of infection. For patients experiencing chronic recurrent uveitis, such as that associated with juvenile idiopathic arthritis, where topical steroid treatments cannot be successfully tapered down, alternative oral, subcutaneous, or intravenous therapies may be utilized. These include oral immunosuppressants, disease-modifying agents, and biologics. However, it should be noted that these systemic medications carry their own risks and potential systemic adverse effects, such as an increased risk of infection and malignancy [7].

Given these existing challenges and limitations in uveitis treatment, biodegradable nano-based drug delivery systems (DDS) present a promising alternative. Their enhanced penetration and superior bioavailability could reduce dosing frequency, thereby improving compliance and minimizing ocular irritation. Furthermore, their potential for controlled release and tissue-specific targeting could mitigate the risk of adverse intraocular pressure increases and cataract formation. Additionally, they may have the capacity to penetrate deeper into the posterior eye segment, possibly eliminating the need for intravitreal injections and their associated risks for the management of posterior segment inflammation. The following section presents a review of the current research state for biodegradable nanosized DDS in uveitis treatment, and Table 3 provides a summary of recent and relevant studies.

### 4.1. Biodegradable Nano-Based DDS for Experimentally Induced Uveitis

In a study conducted by Kasper et al. (2018), they tested the topical treatment of cyclosporin A loaded methoxy-poly(ethylene-glycol)-hexyl substituted poly- (lactic acid) (mPEGhexPLA) nanocarriers (ApidSOL) for effectiveness and tissue distribution in a mouse model of experimental autoimmune uveitis (EAU). The ointment was found to be well tolerated both locally and systemically, and non-toxic when applied to one eye five times a day for nine consecutive days. This resulted in drug accumulation predominantly in the cornea, sclera-choroidal tissue, and lymph nodes of the treated eye, as well as a significantly reduced severity of EAU compared to the untreated counterparts. Furthermore, the treatment regimen was associated with proximally located immunosuppression due to reduced T-cell counts, T-cell proliferation, and IL-2 secretion in the treated eye’s lymph nodes. Overall, the use of cyclosporin A-containing nanocarriers applied topically was found to be an effective treatment for EAU [8].

Other authors have investigated the use of topical treatment, specifically PLGA nanocapsules as eye drops loaded with tacrolimus, to treat ocular inflammation. Rebibo et al. (2021) obtained rats and rabbits to assess the efficacy of a treatment in an EAU model. These nanocapsules were found to be suitable for use in the eye and were able to increase drug retention in the cornea. This also enabled the nanocapsules to penetrate deeper into the eye structures in both porcine ex vivo and rabbit and rat in vivo models, showing improved anti-inflammatory effects compared to the drug in oil solution [9]. With similar success as eyedrops, Chen et al. (2021) reported hydrogel eye drops made from low-deacetylated chitosan and beta-glycerophosphate to deliver adalimumab. The researchers found that the hydrogel eye drops were more effective in treating and delivering the medication than solely using the medication on its own [10].

Another approach to treating uveitis was the use of biodegradable poly(lactic-co-glycolic acid) (PLGA)-based nanoparticles. Luo et al. 2019 conducted a study using biodegradable nanoparticles made of carboxyl-terminated PLGA and divalent zinc ions (DSP-Zn-NP) to encapsulate dexamethasone sodium phosphate. These nanoparticles had high drug content, a small average diameter, and a neutral surface charge. When injected subconjunctivally into the eye, DSP-Zn-NP was deemed to be effective in reducing inflammation in a rat model of autoimmune uveitis for at least 3 weeks. This also decreased the expression of inflammatory cytokines while preserving retinal structure and function and reducing microglial cell density in the retina. Regarding its safety, it was found that four weekly injections of DSP-Zn-NP had similar effects on retinal structure and function as saline injections. Subconjunctival injections of DSP-Zn-NP, importantly, did not result in any retinal toxicity in rats, as assessed by histology and electroretinography, suggesting that DSP-Zn-NP may be a promising and safe treatment option for autoimmune uveitis [11]. During that same year, in 2019, Guo et al. created a nanoparticle-based drug delivery system that was also promising. Specifically, they employed mPEG-PLGA nanoparticles that contained triamcinolone acetonide (TA) using a modified double emulsification method, characterized the loaded nanoparticles, and examined their effects on rats with EAU. They found that the nanoparticles had a spherical shape and were approximately 82 nm in size, and they were also capable of releasing TA for at least 45 days. The TA tended to be incorporated into the hydrophobic PLGA domain of the nanoparticles, while a smaller amount was present in the hydrophilic PEG domain. When tested on the rats, the nanoparticles had stronger anti-inflammatory effects than TA alone, as shown by histopathological examination and changes in the levels of interleukin-17 and IL-10 in the aqueous humor and serum. It should be noted that, given that this is a polymer-based biodegradable drug delivery system, the release of TA from the mPEG-PLGA nanoparticles is primarily influenced by diffusion, swelling, and erosion processes. In this case, the release of TA occurred through diffusion and surface erosion as the mPEG-PLGA polymers degraded. The PLGA copolymer was found to degrade through hydrolysis or biodegradation, breaking down into oligomeric and monomeric constituents [12].

Another study that investigated the use of nanoparticles was by Huang et al. in 2018. They created nanoparticles with a high drug payload for the topical treatment of ocular inflammation. The nanoparticles were created by mixing a succinated triamcinolone acetonide (TA-SA) supramolecular hydrogel with a poly (ethylene glycol)-poly (ɛ-caprolactone)-poly (ethylene glycol) (PECE) aqueous solution. After applying the TA-SA/PECE nanoparticles topically, it was found that they had good ocular biocompatibility and did not cause any irritation or harmful changes. During in vivo rabbit model studies, the nanoparticles also showed better effectiveness at reducing neutrophil infiltration and the quantity of fibrinous exudate in the anterior chamber of the eye compared to treatment with TA alone [13]. Moreover, Xing et al. in 2021 assessed the use of triamcinolone acetonide prepared in poly(D,L-lactide-co-glycolide) (PLGA)-chitosan (PLC) nanoparticles. This formulation was characterized and found to have controlled drug release for 100 h and was biocompatible. In cell and animal studies, the TA-loaded PLC nanoparticles had excellent anti-inflammatory activity and significantly reduced inflammation compared to a control treatment. They also had a longer-lasting pharmacokinetic profile in rabbit eyes compared to control treatments, highlighting the concept that the chitosan coating of this drug delivery system was critical for prolonged drug release [14].

While there have been limited studies on the use of extracellular vesicles (sEVs) for ocular drug delivery, a recent study by Li et al. (2022) examined the effectiveness of using sEVs from mesenchymal stem cells as a drug delivery system for the treatment of EAU using rapamycin as a model drug. The sEVs were loaded with rapamycin and administered to the eye via subconjunctival injection. The study found that the sEVs were able to reach the targeted area in the eye and that the combination of sEVs and rapamycin was more effective at reducing inflammation and cell infiltration than either treatment alone. Additionally, sEVs have the advantage of being naturally occurring, found in all bodily fluids, and able to protect the drugs they engulf from degradation due to their lipid bilayers while overcoming various biological barriers, such as the blood-ocular barrier. Despite the positive remarks about these results, additional research must be performed to thoroughly examine their drug release kinetics and the potential toxic effects of Rapa-sEVs. Still, these results suggest that mesenchymal stem cell-derived sEVs have potential as a drug delivery system for the treatment of EAU and could be an alternative to steroid-based therapies [15]. On a similar note, with respect to sEVs, a study by Garg et al. in 2021 investigated the use of a modified lipid vesicle called a proglycosome nanovesicle from lipid vesicles to improve the effectiveness of tacrolimus in treating EIU in rabbits. This delivery system was found to be deformable and able to sustain the release of tacrolimus for 12 h. The results showed that treatment with these nanovesicles can significantly reduce clinical symptoms of EIU and prevent the breakdown of the blood-aqueous barrier [16].

Another method for the treatment of uveitis is the use of cubosomes. Cubosomes have potential as a new type of ocular drug delivery system due to several advantages. They are made from cost-effective biodegradable lipids using a straightforward and scalable method and are able to incorporate a variety of different types of drugs, including those that are hydrophilic, hydrophobic, and amphiphilic. In another study published in 2020 by Gaballa et al., they proposed the use of cubosomal gels for the delivery of beclomethasone dipropionate for uveitis management while increasing the ocular bioavailability of the drug and increasing its residence time. The results showed that the cubosomes were of a suitable size, had high encapsulation efficiency, and that their use significantly improved transcorneal permeation compared to the control formulation of beclomethasone dipropionate suspension. The optimized cubosomes were incorporated into a gel, which had desirable rheological properties and good ocular tolerability. Importantly, the gel also displayed superior anti-inflammatory properties in the treatment of uveitis in the rabbit model compared to the controls. These findings indicate that cubosomes and the resulting gel may be a promising ocular delivery system for effectively treating uveitis [17].

Antioxidant enzymes, such as SOD1, have the potential to effectively remove reactive oxygen species. However, delivering these enzymes to the eye can be challenging due to their limited penetration. In a recent study by Vaneev et al. published in 2021, they tested a novel treatment option in the form of multilayer nanoparticles made from SOD1, which were deemed to have good stability and a strong therapeutic effect without causing side effects like irritation, acute or chronic toxicity, allergenicity, immunogenicity, or mutagenicity. The researchers tested the nano-SOD1 formulation in a uveitis rabbit model to reduce inflammation. Their results showed that applying nano-SOD1 topically was more effective at decreasing uveitis symptoms, including edema in the cornea and conjunctiva, hyperemia in the iris, and fibrin clots, when compared to the free enzyme. Furthermore, another advantage is that the nano-SOD1 was able to stay on the surface of the cornea better than the native enzyme, which allowed a longer-lasting enzyme and more effective penetration into the interior structures of the eye. Consequently, this not only reduced inflammation but also restored the antioxidant properties of the eye tissue. Therefore, nano-SOD1 may be a promising treatment option for ocular inflammatory disorders [18].

Research by Wu et al. in 2017 developed a dexamethasone sodium phosphate (Dex) supramolecular hydrogel and tested its effectiveness in controlling ocular inflammation. It was found to have thixotropic properties that were influenced by the concentration of calcium ions and Dex. Additionally, the drug release rate from the hydrogel was also dependent on the concentration of calcium ions. When the DDS was administered as a single intravitreal injection in an EAU model in rats, the method was well tolerated with minimal risk of inducing lens opacity and fundus blood vessel tortuosity and showed similar anti-inflammatory effects to the native Dex solution. The hydrogel was found to be effective in reducing inflammatory responses in both the anterior and posterior chambers of the eye while downregulating Th1 and Th17 effector cells. Although the exact molecular processes behind uveitis are not fully understood, there is increasing evidence to suggest that T cells play a role in the development of uveitis [19].

Safwat et al. published an article in 2020 where they looked at using micelles made from a combination of triamcinolone acetonide (TA) and either poly(ethylene glycol)-block-poly(ε-caprolactone) (PEG-b-PCL) or poly(ethylene glycol)-block-poly(lactic acid) (PEG-b-PLA) as a potential treatment for ocular inflammation. Both types of micelles had high drug loading and encapsulation efficiencies, but the PEG2-b-PLA1 micelles had the highest capacity. To extend the drug’s residence time in the eye, the highest-capacity micelles were suspended in chitosan hydrogel. This slowed the drug’s release rate, with only approximately 42% being released after one week, compared to 95% in just 8 h when the drug was suspended on its own. When tested in a rabbit model of ocular inflammation, the PEG2-b-PLA1 micelles suspended in chitosan hydrogel were effective in reducing inflammation and restoring normal corneal tissue [20].

In a study by Tiwari et al. in 2017, they described an ocular self-microemulsifying drug delivery system (SMEDDS) of prednisolone to improve the treatment of experimental uveitis in rabbits. The developed SMEDDS was formulated with linoleic acid, Cremophore RH 40, and propylene glycol, which showed acceptable physicochemical properties, stability, and sustained drug release. When tested topically in a rabbit eye model, it was deemed tolerable without any signs of irritation while showcasing significant improvements in anti-inflammatory activity compared to a marketed formulation. The conclusions of the study suggest that these SMEDDS could potentially be a viable alternative to eye drop treatments due to their ability to increase bioavailability [21]. In a study by Yu et al. published in 2018, they created a hydrogel composed of a GFFY peptide linked to ibuprofen via an ester bond for the treatment of ocular inflammation. The hydrogel demonstrated negligible cytotoxicity and sustained ibuprofen release when activated, showcasing an enzymatically controlled drug delivery method with a higher anti-inflammatory effect when compared to just administering ibuprofen on its own in RAW264.7 macrophages. When this formulation was tested in vivo through a rabbit model, they found that the hydrogel had similar anti-inflammatory therapeutic effects compared to the current treatment, which was sodium diclofenac eyedrops. This was the first study to demonstrate an effective method of supramolecular assemblies in reducing ophthalmic inflammation [22]. However, the biodegradability of the DDS in the two previous studies is uncertain. There is limited evidence on the biodegradability of the self-microemulsifying drug delivery system (SMEDDS), made up of ingredients such as oil and surfactants, used as ophthalmic drug delivery with the peptide supramolecular hydrogel composition.

A noteworthy study utilized a biodegradable intravitreal implant. In a recent research article by Paiva et al. in 2021, they examined the effects of a biodegradable sirolimus-loaded PGLA intravitreal implant for treating uveitis in rabbits. Through clinical and histopathological exams after 35 days, the treated eyes with implants revealed less severe inflammation and reduced damage in cell infiltration in the anterior and posterior portions of the eye while preserving the integrity of blood-ocular barriers. Based on these findings and due to the lack of signs of cataracts, hemorrhage, or other signs of toxicity, the SRL-PLGA implant was deemed to be a safe and promising future treatment for non-infectious uveitis [23].

Finally, Alshmsan et al. evaluated the topical treatment of uveitis through PolyGel™, a poly(α-carboxylate-co-α-benzylcarboxylate-ε-caprolactone)-block-poly(ethylene glycol)-block-poly(α-carboxylate-co-α-benzylcarboxylate-ε-caprolactone) as an in situ gel system for delivery of Cyclosporine A compared to Restasis, and PEO-b-PCL, which is a non-gelling micelle formulation. The irritation studies showed that while PEO-b-PCL and PolyGel are tolerable for use in the eye, the highest ocular bioavailability of cyclosporine A was found in Restasis^®^. The PolyGel allowed for the most prolonged penetration of the drug into the eye, but the amount of the drug that permeated into the eye was initially higher with Restasis^®^ and lower in the other two. Despite this, both CyA-PolyGel™ and Restasis^®^ were effective at reducing inflammation in the eyes of rabbits, making them potentially suitable for treating uveitis [24].

Despite the encouraging results that have been achieved, the full extent of the long-term effects of nano-based biodegradable DDS is still unclear. The studies mentioned above have only used animal models with artificially induced uveitis and have not taken into consideration the effects of DDS on animal models with pre-existing conditions that can cause uveitis. It is crucial that before moving forward with clinical trials, further research is conducted using animal models that have underlying diseases that lead to uveitis, rather than relying solely on those with experimentally induced uveitis. This will provide a clearer understanding of the impact of these DDS on the natural progression of the condition and the potential for any long-term adverse effects.

### 4.2. Biodegradable Nano-Based DDS for Anterior Uveitis

In a study by Wong et al. in 2018, the effectiveness of liposomal triamcinolone acetonide phosphate and liposomal prednisolone phosphate as a treatment for anterior uveitis in rabbits was evaluated. Liposomes have been widely studied for various purposes due to their biocompatibility and biodegradability. In this article, liposomes were administered as a single subconjunctival injection, and the results showed that the liposomal treatment significantly reduced inflammatory scores in the rabbits compared to untreated controls on days 4 and 8 after the induction of uveitis. The liposomal treatment was also found to be more effective at reducing inflammation than topical prednisolone on day 8. Furthermore, the anti-inflammatory effect of the liposomal treatment persisted even after being challenged with the antigen on day 11. Through histology and immunostaining, the localization of the liposomes was observed to remain in the eye for at least one month. In addition, the researchers highlighted that this methodology poses little risk of globe injury compared to peribulbar injections and provides no risk of endophthalmitis, which is usually associated with intravitreal and intracameral injections. However, this study was not designed to assess the potential adverse effects of the treatment, and the follow-up period may not have been long enough to identify the development of long-term complications such as cataracts and increased intraocular pressure. This study was the first to compare the effectiveness of a single subconjunctival injection of liposomal steroids to a single injection of unencapsulated steroids and to the current standard treatment of intensive topically applied steroid eyedrops in treating anterior uveitis [25].

Another lipid carrier was assessed according to a study by Garg et al. published in 2021. Cationic nanostructured lipid carriers (NLCs) were prepared as a drug delivery system to increase the penetration and retention of corticosteroids in the eye. These NLCs, which contain the drug triamcinolone acetonide, were small in size (less than two hundred nanometers), contained a positive charge, and were made using a hot microemulsion method. They released the drug slowly and sustainably over a period of 24 h, and it was shown to be non-toxic, non-irritant, and effective in reducing inflammation in cells. The cationic NLCs also had a high level of drug entrapment efficiency (88%) and could be taken up by cells, remaining inside for a period of 24 h but penetrating deeper into eye layers within 2 h. These characteristics make cationic NLCs a promising option for improving the effectiveness of corticosteroid treatment for anterior uveitis and other ocular conditions [26].

In addition, the study by Alami-Milani et al. (2019) revealed that polycaprolactone-polyethylene glycol-polycaprolactone (PCL-PEG-PCL) micelles could be used to improve the anti-inflammatory effects of dexamethasone (DEX) in the treatment of anterior uveitis. They showed that the PCL-PEG-PCL micelles had good compatibility and uptake by cells, and they reduced the symptoms of uveitis after a lag time. However, the micelles were not significantly more effective than the marketed dexamethasone eye drop at 24 and 36 h after treatment. This suggests that the PCL-PEG-PCL micelles have potential as carriers for DEX in the treatment of uveitis, but more research is needed to fully understand their potential, such as including more trials and monitoring the prolonged anti-inflammatory impacts of sustained-release of the compound [27].

Moreover, an important development in the field of anterior uveitis treatment is the use of biodegradable hydrogels. In a study by Fang et al. in 2022, polypseudorotaxane hydrogels were created by mixing Soluplus micelles with a cyclodextrin solution. These hydrogels have the ability to thin under shear force and release their contents over an extended period of time. They also showed higher transcorneal permeability, increased precorneal retention, and intraocular bioavailability of flurbiprofen in animal studies. These hydrogels were also effective at reducing inflammation in a rabbit model of endotoxin-induced uveitis with fewer administrations and were shown to be safe in cytotoxicity and ocular irritation studies [28].

In another article by Yu et al. in 2019, they developed and tested dexamethasone-peptide conjugate, which is made up of a combination of dexamethasone and a peptide connected by a biodegradable ester bond. The formulation could form a high concentration of nanoparticles in aqueous solution to treat anterior uveitis, and topical instilled application was well tolerated as it did not cause any significant side effects such as alteration of the thickness of the cornea or intraocular pressure. This new formulation was found to be just as effective as aqueous solutions containing dexamethasone sodium phosphate [29].

### 4.3. Biodegradable Nano-Based DDS for Posterior Uveitis

Posterior uveitis is a difficult to manage condition due to its localized inflammation in the posterior segments of the eye and the presence of critical structures such as the macula, optic nerve, and retinal vessels. Irreversible vision loss and blindness can quickly occur if these critical structures are affected. Diagnosis is more challenging than anterior uveitis, as microbiological sampling is difficult and the condition is often associated with underlying infectious or systemic causes that require multimodal treatments. Invasive administration routes, such as periocular or intravitreal injection, are often necessary as topical eye drops cannot effectively penetrate the ocular barriers to reach the posterior segment. Systemic medications may also be used in bilateral cases but have poor penetration and bioavailability to ocular structures due to the blood-retinal barrier, as described in the first section of this article.

In the literature, it is known that both intravitreal and periocular injections of triamcinolone acetonide suspension maintain a high risk for negative effects such as high intraocular pressure and retinal toxicity, despite being a treatment option for non-infectious posterior uveitis. Therefore, in 2018, Xiong et al. reported the use of hydrogels as an alternative treatment. They generated an injectable glycosylated triamcinolone acetonide hydrogelator (TA-SA-Glu) hydrogel that is thermosensitive to ease uveitis. This novel biodegradable DDS was found to have minimal retinal toxicity when injected at a dosage of 69 nmol per eye in an in vivo rat study and was more effective in controlling non-infectious posterior uveitis than the conventional TA injection. Particularly, the TA-SA-glu hydrogel system aids the downregulation of pro-inflammatory effector responses in Th1 and Th17 effector cells [30].

In 2021, Mehra et al. proposed the use of a topical nanomicellar formulation using Soluplus, a copolymer of polyvinyl caprolactam-polyvinylalcohol-polyethyleneglycol (PVCL-PVA-PEG) to deliver everolimus to treat posterior uveitis using ex vivo goat cornea and in vitro methods. The nanomicelles were found to have a low critical micelle concentration, be 65.55 nm in size, and have a smooth surface, as well as high encapsulation efficiency and sustained release of everolimus. They also showed significantly higher permeation across the goat cornea compared to everolimus suspension and deeper permeation through the cornea, as confirmed by confocal laser scanning microscopy. The nanomicellar formulation also improved drug bioavailability while remaining in the circulatory system for a longer duration and seeming to have accumulated in the inflammatory site of interest. As this drug delivery system demonstrated adequate stability, there was no toxicity in the eye as found in the Hen’s egg test-chorioallatonic membrane assay, so it can be reassured that the prepared method is safe for ocular use [31].

Nanoparticles have also been researched for posterior uveitis. In a study by Badr et al. in 2022, they employed a mouse model of posterior EAU to assess its treatment with rapamycin in a nanoparticle-based eye drop. The compound called Molecular Envelope Technology-Rapamycin (MET-RAP) was successful in controlling the progression of the disease by reducing the level of a protein called RORγt, increasing the expression of Foxp3, and increasing the secretion of IL-10. These effects likely play a role in the mechanism that shifts the balance between T helper 17 cells and regulatory T cells, which in turn reduces the progression of EAU. Based on these results, MET-RAP eye drops may be a promising treatment for retinal inflammatory diseases [32].

### 4.4. Biodegradable Nano-Based DDS for Endophthalmitis

Endophthalmitis is an inflammation of the vitreous and aqueous fluids within the eye caused by a bacterial or fungal infection. It can occur due to various reasons, such as intraocular surgery, trauma, or corneal ulcers. The hallmark of endophthalmitis is the infiltration of the vitreous cavity and anterior chamber with inflammatory cells, leading to clinical signs of vitritis and hypoplasia. Treatment options can include eyedrops or intravitreal methods containing antibiotics or antifungals with multiple drug combinations, and pars plana vitrectomy (PPV). However, the use of intravitreal antibiotics can result in side effects such as retinal toxicity and corneal opacification. PPV is an invasive surgical procedure that comes with several risks and complications, including but not limited to retinal tears and detachments, increased intraocular pressure, re-infection, and permanent vision loss. Systemic antibiotics are usually not sufficient as monotherapy and are used in combination with intravitreal antibiotics [33].

A study by Coburn et al., published in 2019, found that biomimetic nanosponges, synthetic materials that mimic the properties of red blood cells, have been shown to neutralize pore-forming toxins and preserve retinal function. Their findings showed that nanosponge pretreatment reduced hemolytic activity in vitro, improved retinal function, and reduced ocular pathology in a murine model of endophthalmitis. Treatment with gatifloxacin and gatifloxacin-nanosponges also reduced intraocular bacterial burdens and decreased ocular pathology and inflammation. In the future, research should be conducted to determine the pharmacokinetics of red blood cell-derived nanosponges when used in combination with antibiotics to treat eye infections, as well as to optimize the concentrations and timing of administration. This will provide the necessary foundation for the use of nanosponges as a supplementary treatment for bacterial intraocular infections [34]. While nanosponges showcase promising data, unilamellar liposomes have also recently been investigated for periocular transscleral co-delivery of steroids and antibiotics, offering potential new possibilities for treating chronic ophthalmic infections [35].

Alternatively, nanoparticles can be useful for the management of endophthalmitis. An article by Mahaling et al. in 2021 created a nanoparticle delivery system made up of a hydrophobic polylactic acid (PLA) core and a hydrophilic chitosan (CHI) shell, which contained either the antibiotic azithromycin (AZM) or the corticosteroid triamcinolone acetonide (TCA). These nanoparticles were developed for the treatment of endophthalmitis and were administered as an eyedrop. The delivery system showed good compatibility with blood and tissues as well as sustained release of the drugs, making it suitable for long-term treatment of endophthalmitis. The combination of PLA-CHI-AZM and PLA-CHI-TCA nanoparticles was more effective in treating endophthalmitis than any of the individual components alone, due to their enhanced antibacterial and anti-inflammatory properties. This nanoparticle-based combinatorial drug delivery system administered as an eye drop could be a promising non-invasive therapy for endophthalmitis [36].

### 4.5. Biodegradable Nano-Based DDS for Postoperative Uveitis and Endophthalmitis

Receptor-mediated drug delivery has been recently explored for the use of postoperative uveitis treatment. In a recent study published in 2020, Ganugula et al. assessed the capacity of receptor-mediated delivery of curcumin to decrease inflammation in a model of lens-induced uveitis. The researchers successfully encapsulated curcumin in double-headed polyester nanoparticles using gambogic acid as a coupling agent and PLGA as the polymer. When administered orally to canine models with lens-induced uveitis, these PLGA-GA2-CUR nanoparticles resulted in significant levels of curcumin in the aqueous humor and produced comparable clinical effects to commonly used anti-inflammatory medications. This novel nanoparticle delivery system may enhance the bioavailability of curcumin while reducing the adverse effects associated with the use of topical corticosteroids or NSAIDs [37].

Another method that has been tested and proven successful is the use of a chitosan-based hydrogel, as reported by Cheng et al. in 2019, for the treatment and prevention of postoperative endophthalmitis. Their goal was to generate sustained drug release that could deliver levofloxacin safely and effectively without the need for injections or topical eyedrops. When tested in the laboratory, in vitro studies showed that the hydrogel was found to release the drug in a sustained manner and showed good antibacterial properties against certain bacteria through the observation of a significant inhibition zone of the bacteria and the long-term antibacterial property of the developed hydrogel. In addition, the hydrogel was found to be biocompatible with corneal epithelial cells [38]. The following year, an article by Cheng et al. 2021 created and studied a dual drug delivery system known as PAgel-LNPs, which is made up of a thermosensitive chitosan/gelatin-based hydrogel that is able to sustainably release two drugs, prednisolone acetate and levofloxacin-loaded nanoparticles (LNPs). They suggest that the optimal concentrations of these drugs for the treatment of corneal epithelial cells are 5 μg/mL and 50 μg/mL for LNP and prednisolone acetate, respectively. PAgel-LNPs contain a porous structure and have been shown to be biocompatible, thermosensitive, and capable of sustained drug release. In laboratory tests using damaged corneal epithelial cells and a rabbit model of S. aureus keratitis, the system demonstrated anti-inflammatory and anti-bacterial properties. This drug delivery system may have the potential to be used for the treatment and prevention of postoperative uveitis and may also improve patient compliance due to its ability to sustainably release both levofloxacin and prednisolone acetate [39].

Furthermore, unique nanoparticles composed of AuAgCu_2_O-bromfenac sodium (AuAgCu_2_O-BS NPs) have shown promising results. In an article by Ye et al. published in 2020, they formed AuAgCu_2_O-BS nanoparticles that were designed to combine anti-bacterial and anti-inflammatory effects to treat postoperative endophthalmitis after cataract surgery. The preliminary toxicity investigations of the nanosystem revealed its superior biocompatibility, with low levels of cytotoxicity and minimal impact on intraocular pressure and other major organs. This study provides a promising synergic therapeutic strategy for the treatment of post-cataract extraction endophthalmitis [40].

## 5. Overview of Biodegradable Ocular Implants (Ozurdex) for Uveitis

Numerous ocular diseases can be treated using a range of commercially available implants that are approved by the FDA. Some of these implants that are biodegradable and bioresorbable include Iluvien^®^, Ozurdex^®^, and Dexycu^®^. Ozurdex, which is composed of a copolymer of lactic and glycolic acids containing micronized dexamethasone, is primarily used for the treatment of uveitis. Several recent studies in the literature have shown that Ozurdex can effectively and safely treat various forms of uveitis, with minimal adverse effects observed in clinical trials. Furthermore, other studies have also explored the possibility of replacing systemic corticosteroid therapy with Ozurdex to avoid the unpleasant side effects associated with high doses of corticosteroids. Refer to Table 4 for additional details.

## 6. Biodegradable Nano-Based DDS for Neuro-Ophthalmologic Diseases and Retinal Ganglion Cell Death

Optic neuropathy is a condition in which there is damage to the optic nerve, a bundle of nerve fibers in the retina that sends visual signals to the brain. Optic neuropathy has various causes, including but not limited to ischemic, inflammatory, infiltrative, traumatic, compressive, metabolic, and glaucomatous types. The treatment of optic neuropathy depends on its etiology, and in many cases, it is a late complication of the underlying disease that cannot be reversed as the lost retinal ganglion cells cannot be regenerated with current conventional treatments [57].

The development of nano-based drug delivery systems (DDS) is underway for treating optic neuropathy. These DDS have the potential to deliver neuroprotective agents such as neurotrophins (NTs) and oxidative stress modulators, as well as gene therapies. Neurotrophins are particularly attractive due to their crucial role in maintaining neuronal health and survival [58]. The treatment of neuro-ophthalmologic diseases faces challenges due to the difficulty in delivering these drugs to the posterior segment of the eye and neuronal cell layers, the short half-life and rapid clearance of drugs, and the requirement for multiple injections with associated systemic side effects [59]. To address these challenges, various nanocarriers have been explored for targeted drug delivery to the inner retina and retinal ganglion cell (RGC) layer related to optic nerve damage. This targeted approach has demonstrated 100-fold or higher drug concentrations at focal sites compared to systemic delivery, reducing off-target effects, overcoming clearance mechanisms and physiological barriers, and protecting the drug from degradation [60]. Table 5 presents a concise summary of recent, relevant studies.

### 6.1. Biodegradable Nano-Based DDS for Traumatic Optic Neuropathy

In preclinical studies, several nano-based drug delivery systems (DDS) have been investigated for their ability to target retinal ganglion cell degeneration in traumatic optic neuropathy (TON). In one study by Wang et al. (2020), ciliary neurotrophic factor (CNTF) was co-delivered with the antibiotic FK506 in a chitosan-based thermosensitive hydrogel for localized targeting of RGCs in a rabbit model of TON [59]. FK506 was introduced into the CNTF-hydrogel following its loading into a polyethylene glycol-block-poly (benzyl glutamate) copolymer (PEG-PBG) nanomicelle. Chitosan is a well-established non-toxic biopolymer that is degraded in the body by lysozyme-based enzymatic hydrolysis [72]. Co-polymers of chitosan and PEG have also been widely used for their low solubility and high biodegradability [73]. Concerns remain for the true biodegradability of polyethers such as PEG and PBG; however, current consensus states that molecules with molecular weights >20,000 Da show biodegradability through oxidative means [74]. The study’s results showed that the disordered RGC layer observed after injury was largely reversed, with fewer RGC deaths following treatment, and that the hydrogel was protective against axonal damage. It should be noted that the drug administration was carried out by smearing the gel at the injury site rather than through traditional forms of clinical administration. When examining TON in rat models, further information has been gleaned on the mechanisms underlying RGC damage as well as the feasibility of nano-based DDS. Through the delivery of deferoxamine-loaded hyaluronic acid (HA) nanoparticles, Lin et al. (2022) validated the previously hypothesized role of ferroptosis in inducing RGC death and demonstrated RGC protection through suppression of ferroptosis by the iron chelator deferoxamine [61]. HA is an extremely promising candidate, particularly in ocular nanotechnology, due to its natural presence and prominence in the eye. HA eye drops are common therapies worldwide, easing the process of introducing HA in novel therapies as well [75]. In a similar study, Wang et al. (2021) delivered the neuropeptide PACAP38 to the RGC layer within RGC-derived exosomes [62]. This delivery vehicle possesses the benefit of being a naturally derived DDS with innate entry and removal mechanisms that overcome the inflammatory side effects of alternate routes of entry. The drug specifically localizes to the target of interest by employing exosomes from that same source. This is what Wang et al. (2021) observed, as the exosome-loaded PACAP (EXO*_PACAP38_*) showed greater RGC uptake and subsequently enhanced RGC survival as well as preservation of the thickness of the retinal neural fiber layer [62]. Neurodegeneration arises due to complex interactions between the neuronal system, its surroundings, and downstream damages. Thus, targeting downstream mediators of neuronal damage is an appealing avenue for treatment. Maxwell et al. (2021) developed injectable alginate hydrogels containing Methylene Blue (MB) to target the production of reactive oxygen species (ROS) contributing to RGC death in TON [63]. They observed a prolonged release of MB for up to 12 days compared to free MB. ROS levels decreased significantly in in vitro cell models and were correlated to but not directly assessed in neuroprotection. Likewise, no work was conducted in vivo to corroborate these findings. Additionally, the impact of in vivo fatty tissue on the release of MB and the optimal local dosing of MB remain unknown, highlighting the need for further in vivo experiments. While biodegradable, alginate hydrogels present some challenges as ion exchange is required to uncross link the polymers. Further research is necessary to better understand the efficiency, rate, and safety of hydrogel breakdown in the ocular system [62].

### 6.2. Biodegradable Nano-Based DDS for Retinal Ganglion Cell Degeneration and Oxidative Stress

A recent study by Lou et al. (2021) explored the use of biodegradable polydopamine (PDA) nanoparticles following RGC degeneration in optic nerve injury [64]. As with Maxwell et al. (2021), the goal was to target ROS production to induce RGC survival. The PDA nanoparticles alone showed promising RGC protection through the elimination of several types of ROS. PDA also had a positive effect on axon regeneration. With combined delivery of brimonidine, an approved medication for glaucoma, greater RGC density and survival were observed compared to treatment with brimonidine or PDA alone, supporting a synergistic effect. This highlights a key benefit of nano-based DDS: allowing for the delivery of multiple drugs targeting the same or different components of disease to provide a more holistic effect with a single treatment. Brimonidine typically has poor drug bioavailability and rapid clearance, which could be overcome by delivering this drug in PDA nanoparticles. In another study where RGC loss was induced by oxidative stress, Giannaccini et al. (2018) delivered nerve growth factor (NGF) and brain derived neurotrophic factor (BDNF) conjugated to magnetic nanoparticles. Magnetic nanoparticles (MNPs) are an exciting DDS for ocular disease due to their innate self-localization in the retina, which shows promise in overcoming the challenges associated with barrier penetration in ocular DDS. The degradation of MNPs in vivo does depend on the coating molecules and instigates changes in the magnetic properties of the degraded particles, which are important considerations when designing and optimizing MNP-based DDS [76]. Nevertheless, the study revealed another promising approach to taking advantage of the neuroprotective effects of NTs. The conjugated proteins remained stable for 2 weeks and completely prevented RGC loss, whereas free NTs had no effect in zebrafish larvae. Another appealing consideration in nano-based DDS is the use of nanotechnology to enhance topical drug administration, which serves as a non-invasive and patient-friendly therapy. Because neuro-ophthalmologic diseases primarily affect the posterior eye and inner retina, drug delivery is rather difficult. Particularly with topical treatment, the distance the drug must cover and the barriers it must overcome severely limit the success of ongoing trials. Nanotechnology is one promising approach, and on this front, Bessone et al. (2020) assessed the effects of an ethylcellulose nanoparticle in the topical application of melatonin, a circadian hormone with previously reported neuroprotective effects on RGCs [66]. In a rabbit model of retinal degeneration, topical application of nanoparticle-loaded melatonin demonstrated great trans-corneal permeation and a slow, stable release compared to melatonin alone. This was linked with the maintenance of the neuronal layer thickness and organization as well as RGC survival, which was not observed at all when melatonin was administered alone. Similar results were achieved by Davis et al. (2018) when curcumin-loaded succinate nanoparticles were topically administered twice daily for three weeks, after which a significant reduction in RGC loss was observed [67].

### 6.3. Biodegradable Nano-Based DDS for Optic Nerve Crush Models

Several studies have explored nano-based DDS for general optic nerve damage through well-established optic nerve crush (ONC) models. Such studies have further advanced our understanding of the types of therapies that can be delivered using nanotechnology. Of great interest is the non-viral delivery of gene therapies by alternatively employing nanoparticles. In one such study, Tawfik et al. (2021) successfully selectively silenced caspase-3 through the delivery of caspase-3 small interfering RNA in polybutylcyanoacrylate nanoparticles [68]. Pure siRNA delivery has inherent risks, including rapid degradation by ribonucleases, a short life span, and the need for extensive engineering [67]. Treatment following ONC showed a significant blocking of caspase-3 expression both in vitro and in vivo in rats, as well as an almost 25% recovery of RGCs. Interestingly, Huang et al. (2021) found that there was significantly improved siRNA delivery and DDS localization to the inner retina and RGCs when lipid nanoparticles of stronger positive charge were used compared to moderately positive or negative nanoparticles [69]. However, sufficient drug delivery seems to occur with corresponding improvement even when negatively charged DDS are used [65]. These results nonetheless shed light on the importance of the DDS formulation and its individual components for localization to desired cell layers and target tissues. In another ONC mouse model, Sung et al. (2020) observed successful delivery of histone deacetylase protein, trichostatin A, to the RGC layer of the inner retina with positive PEGylated liposomes, resulting in increased RGC survival and reduced apoptosis [77]. Conventional liposomes are still rapidly degraded, and thus the stabilization of these liposomes with polymers like PEG accounts for improved, longer-term drug delivery [78]. However, liposome PEGylation does hinder the hydrolytic degradation of the system within the body, and thus it is an important factor to consider when formulating these delivery molecules [77].

### 6.4. Biodegradable Nano-Based DDS for Other Neuro-Ophthalmologic Diseases

In an optimization study for PLGA microspheres loaded with idebenone for the treatment of Leber’s hereditary optic neuropathy (LHON), Varela-Fernández et al. (2022) observed site-specific maintained release with no cytotoxicity [70]. PLGA is an FDA-approved polymer that is easily degraded into biocompatible products that can be excreted by normal physiological pathways [78]. In vitro studies possess the benefit of allowing for the assessment of a disease through various animal models and routes of disease induction. With retinal degeneration, disease can be induced through the ONC model or oxidative stress, as previously discussed. In another study, Eriksen et al. (2018) induced RGC death through excessive delivery of NMDA in mice, following which the effectiveness of several targets of the mammalian target of rapamycin signaling pathway, including CNTF, was assessed when delivered in a liposomal system [71]. It was generally observed that axon regeneration and RGC survival improved following treatment compared to controls. RGC survival was further induced when liposome delivery was combined with RGC transplantation, which alone was not able to prevent RGC loss completely. This opens the doors to using nano-based DDS synergistically with other therapies as well to maximize therapeutic potential. Likewise, customization of the nanotechnology to the drug and disease of interest opens the door to several potential therapies that were not possible before. Erikson et al. (2018) achieved simultaneous delivery of various drugs due to the dual hydrophobic-hydrophilic nature of the cholesterol liposomes used, where both lipophilic and hydrophilic molecules can be loaded within the same system for delivery [70]. CNTF was again delivered to the inner retina via nanoparticles in a study by Yang et al. (2021), who observed pro-survival and pro-proliferative effects in vitro and, as expected, improved RGC survival and vision preservation in vivo for up to 70 days [69]. This was the longest timeline of any of the included studies, supporting the stability and longevity of nano-based DDS in ocular disease. They further examined the effect of loaded drug compared to free drug in vitro and found no difference, concluding that nanoparticle loading does not hinder the drug’s efficacy.

There are several considerations to be taken into account when assessing the success of the nano-based DDS studies discussed above. Firstly, the studies were carried out in preclinical animal models or cell cultures, which may not accurately reflect human conditions due to the complexity of human biology and the presence of comorbidities. Additionally, in these preclinical studies, treatment is usually administered immediately after injury. However, in a clinical setting, many neuro-ophthalmologic diseases may occur before symptoms appear for diagnosis, leading to a delay in treatment initiation, often several months to years after the onset of retinal ganglion cell degeneration. It is also well established that secondary pathological modifications occur as neuronal and cellular degeneration takes place [67]. Thus, further work should be conducted to assess the effectiveness of nano-based DDS in later stages of the disease.

## 7. Challenges, Future Perspectives, and Future Research Directions

Various ocular diseases are on the rise, with more and more patients affected each year, and they are predicted to continue to rise in the near future. This calls for rapid advancement in successful clinically approved treatments that can combat disease progression in the long term [2,79]. There is promising data in the rapidly expanding field of nano-base biodegradable DDS for uveitis and neuro-ophthalmologic conditions, highlighting their potential advantages over conventional methods. However, although significant progress has been made, there is much more to learn before its translation into clinical testing—the vast majority of studies are still in the preclinical phase. A foreseeable hurdle is the accurate replication of human physiological features and overcoming anatomical differences in animal model studies [80]. As a result, despite significant efforts to develop nano-based ocular delivery systems, minimal success has been achieved in the clinical phase, which is necessary to transition the technology into the market. The promising results obtained in animal models have yet to show successful replication in humans, likely due to differences in physiology, including the size difference between animal models and humans, the complex multifactorial nature of the diseases, and the delayed diagnosis of many diseases in humans, which leads to secondary complications [81]. Rabbits are commonly used for ophthalmic research due to the comparable size of the eye compared to rodent models; however, significant physiological and anatomical differences still exist, which hinder in vivo success in humans. Primate models are more appealing on this front; however, there is limited research working with primate models due to higher costs and a lower number of animals available, which limits large-scale studies [81]. Additional research is required to fully elucidate the long-term safety profile, sustained effectiveness, dosage regimens, frequency of administration, and more. Despite these limitations, there remains a positive outlook for biodegradable nano-based DDS for ocular therapy in the near future.

With respect to biodegradable intravitreal implants like Ozurdex, they represent one avenue for treating uveitis; however, their use may pose some limitations and risks. The administration of Ozurdex has been associated with an increase in intraocular pressure within the first two weeks of injection, reaching its peak at approximately sixty days post-injection [82]. While there is no known correlation between the progression of glaucoma and the number of DEX implants, repeated injections may increase the risk of cataract progression [82,83]. Other associated risks include floaters and conjunctival hemorrhage. Additionally, a higher-than-expected prevalence of endophthalmitis and retinal detachment has been reported with the use of intravitreal DEX implants, possibly due to the implantation technique or the DEX implant itself. The larger bore of the 0.413 mm (22-gauge) DEX implantation device, when compared to 0.159 mm (30-gauge) intraocular needles, may create a larger needle tract in the globe, thereby increasing the chances of developing endophthalmitis [84]. Finally, it is important to note that the effect of intravitreal Ozurdex implants is only limited to a duration of 3–6 months [85]. Further research is needed to evaluate its long-term efficacy, including longer follow-up assessments and larger sample sizes. Looking towards the future, Re-Vana Therapeutics Ltd. is currently in the process of developing OcuLief™ and EyeLief™, which are photosensitive biodegradable implants that could potentially revolutionize and shape the future of ocular drug delivery systems [86].

Significant attention has been focused on developing non-invasive and sustained drug releases for eye disorders. Repeated injections, the current gold standard for several ocular conditions, face several challenges, including patient compliance, large healthcare and personal costs, and damage to ocular structures such as retinal detachment [87]. Minimizing the need for frequent injections will help overcome these limitations. Sustained delivery is especially important for degenerative disorders or cell damage resulting in vision impairment, which currently leads to irreversible damage in the long term. Sustained cell-based and tissue engineering therapies are thus a primary focus [79]. The other major focus in nano-based delivery systems is the transparency of the biomaterials, a prerequisite for ophthalmologic applications [88]. This is also closely linked to the development of biodegradable delivery systems, for which the time of degradation, rate and route of clearance, and toxicity are important considerations currently being explored. Natural biopolymers have shown more success on this front as they best replicate native ocular structures; however, they do present the challenge of limited modifiability and tenability.

Despite these limitations, significant efforts are being made to tackle ocular diseases with various biomaterials and delivery options, and in recent years, much progress has been made to enhance the replicability of natural structures, prolong therapeutic efficacy, and enhance biocompatibility while maximizing patient comfort.

## 8. Conclusions

In conclusion, our comprehensive review of recent advancements in biodegradable nano-based drug delivery systems (DDS) for uveitis and neuro-ophthalmologic conditions underscores the potential of this innovative approach. The ocular barriers, which have long limited the effectiveness of traditional therapies, can be successfully navigated by these nanoscale systems, thus significantly improving drug bioavailability and ocular tissue residence time. Moreover, the inherent biodegradability of these polymers minimizes adverse reactions and toxicity, further enhancing the clinical applicability of these systems. The years 2017 through 2023 have seen a surge in preclinical and clinical studies, all collectively showcasing the rapid evolution and potential of nano-based DDS in revolutionizing the treatment strategies for these disorders.

Moving forward, continued advancements in biopolymer science and a deeper understanding of ocular pharmacology will undoubtedly lead to the refinement of current DDS designs and the development of even more effective systems. Our review has painted an encouraging picture of the progress made thus far and the potential for further groundbreaking discoveries in the field of biodegradable nano-based DDS for ocular conditions.

Thus, the future of managing ophthalmic conditions seems bright with the continued development and implementation of biodegradable nano-based DDS. As we continue to navigate this exciting frontier of ocular medicine, it is our hope that these advancements will translate into improved patient outcomes, opening a new chapter in the effective management of these challenging conditions.

## Figures and Tables

**Figure 1 pharmaceutics-15-01952-f001:**
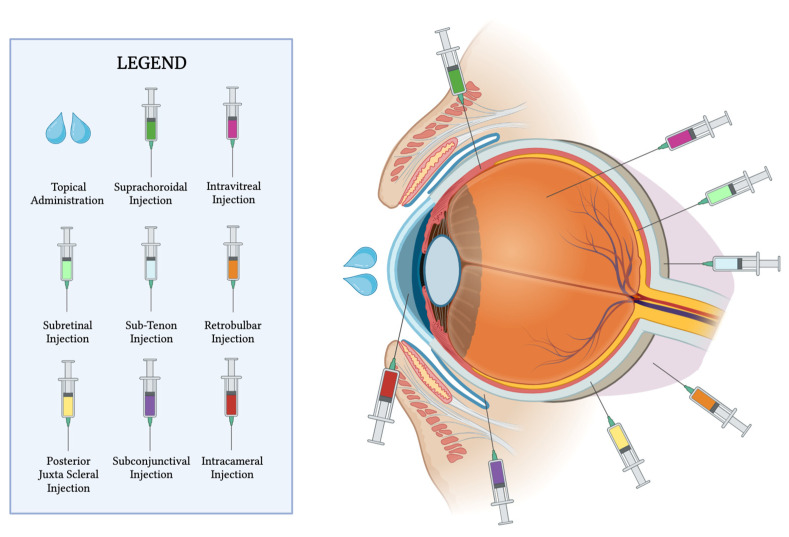
Diverse routes are available for ophthalmic medication delivery. This figure offers an overview of the different administration methods, encompassing topical, subconjunctival, suprachoroidal, intracameral, intravitreal, retrobulbar, sub-tenon, posterior juxta-scleral, and subretinal approaches.

**Figure 2 pharmaceutics-15-01952-f002:**
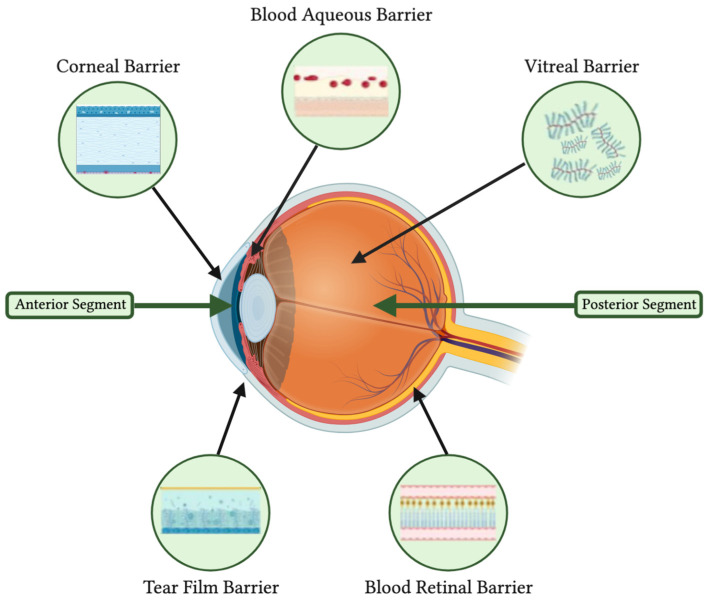
Anatomical barriers in ocular drug delivery.

**Figure 3 pharmaceutics-15-01952-f003:**
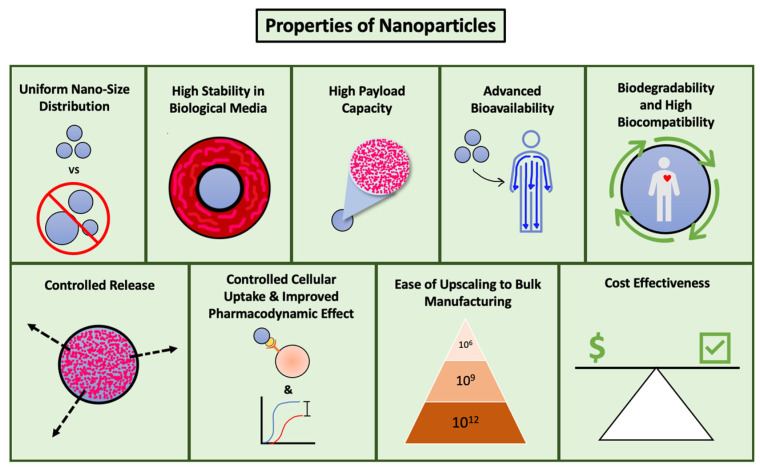
Ideal properties of nanocarriers.

**Table 1 pharmaceutics-15-01952-t001:** Overview of the characteristics and advantages of biopolymers.

Biopolymer	Characteristics	Advantages
Hyaluronic Acid	Anionic polymer and high water-retention capacity.	Can be used in hydrogel formulations. Prolonged tear film stabilization. Can coat liposomes to elongate drug release and increase cellular uptake.
Cellulose	Can self-assemble into nanorods, nanospheres, nanosponges, and nanorods upon functionalization with a copolymer, allowing for ease of bulk manufacturing.	Used in conventional eye drops to improve viscosity. Different derivatives are extensively applied in ocular drug delivery. HPMC swells in contact with water, forming hydrogels with mucoadhesive properties.
Chitosan	Requires chemical modification, is mucoadhesive, and has unique in situ gelling properties.	Excellent corneal permeation. Can temporarily open tight junctions to increase drug permeability. Can increase antibacterial activity when functionalized with quaternary ammoniums.
Alginate	Anionic copolymer that can exert cell immobilization and can be used in copolymeric nanoparticles with chitosan derivatives.	Suitable polymer in hydrogels. Terminable hydrogels due to reversible gelation.
PLGA	Commonly used, subject to abundant modifications, can be enhanced in size and surface potential, and can be modified with PEG.	Sustained drug release. High entrapment efficiency. Topical administration can deliver drugs to the posterior segment of the eye.
Poloxamers	Biodegradable, mucomimetic, and non-ionic surfactants.	Approved by the FDA as a vehicle for ocular drug delivery. Used in various ocular formulations.
Cyclodextrins	Cyclic oligosaccharides can form hydrophobic cavities with externally hydrophilic surfaces.	Can increase the bioavailability of many guests. Wide range of formulations with great potential for ocular drug delivery.

FDA: Food and Drug Administration. PEG: Poly(ethylene glycol). PLGA: Poly(lactide-co-glycolide).

**Table 2 pharmaceutics-15-01952-t002:** Comparative summary of drug delivery systems for ocular applications: key characteristics and benefits.

Drug Delivery System	Characteristics	Advantages
Nanomicelles	Spherical structures made up of surfactant molecules that self-assemble in water or polar solvents.	Ideal for encapsulating hydrophobic compounds. Enhanced ocular micro-adhesion. Increased efficiency with size reduction. Potential for stimulus–responsive release. Higher encapsulation capacity with increased amphiphilic polymers.
Liposomes	Vesicles composed of one or more phospholipid bilayers.	Flexible for chemical modification. Improved pre-corneal and conjunctival penetration. Enhanced bioactivity. Altered head group charge for selective interaction with mucin and corneal permeability. PAMAM-coated liposomes for better corneal barrier penetration.
Dispersed nanoparticles	Self-assembling supramolecular assemblies.	Increased carrying efficiency. Selective tissue accumulation. Controlled drug release. Various architectures are available. Enhanced permeability and retention effect. Mucoadhesive characteristics.
Dendrimers	Repeating multibranched polymers with high-density functional groups.	High encapsulation efficiency. Predictable biodistribution. Customizable hydrogels. Superior loading capacity. Narrow polydispersity. Can serve as building blocks for nanogels or liposomes.
Hydrogels	Highly absorbent polymer networks.	pH and thermoresponsive. Extended drug release. Injectable. Potential for targeted drug delivery. Ultrasound-responsive. Reduced frequency of intravitreal injections.
Nanosuspensions and nanoemulsions	Aqueous dispersions of insoluble drug particles or droplets of one liquid in another liquid.	Improved solubility and stability of poorly soluble drugs. Enhanced bioavailability. Prolonged drug release profiles. Stabilized with amphiphilic salts of cholesterol. Increased viscosity and retention time with dispersed oil phases and water-soluble polymers.
Microneedles	Small, needle-like structures.	Self-administration. Accurate drug delivery to the target site. Fewer complications compared to traditional injections. Hollow microneedles for greater guest molecule loading.

**Table 3 pharmaceutics-15-01952-t003:** Biodegradable nano-based DDS for uveitis.

Disease	Drug	DDS	Advantages and Considerations	Administration Route	Stage	Reference
EAU	Cyclosporin A	mPEGhexPLA nanocarriers	Well tolerated locally. No immediate toxicity after repeated application. Applied noninvasively as eye drops. Versatile: nanocarrier suitable for topical/systemic application. Known to deliver other drugs.	Topical	Preclinical mouse models in vitro and in vivo.	[8]
EAU	Tacrolimus	PLGA nanocapsule	Homogeneous size. High encapsulation efficiency. No eye irritation after multiple applications.	Topical	Preclinical rabbit models ex vivo and in vivo.	[9]
EAU	Adalimumab	Low-deacetylated chitosan and β-glycerophosphate hydrogel	No significant adverse effects. Higher drug release. Amounts within limited time. Excellent therapeutic efficacy.	Topical	Preclinical rat models in vitro and in vitro.	[10]
EAU	Dexamethasone sodium phosphate	Carboxyl-terminated PLGA with DSP-Zn-NP	Improved loading and sustained release of dexamethasone. Reduced inflammatory cell infiltration. Preserved retinal function. No retinal toxicity.	Subconjunctival injection	Preclinic rat models ex vivo and in vivo.	[11]
EAU	Triamcinolone acetonide	mPEG-PLGA nanoparticles	Could sustain for more than 45 days. Good biocompatibility. High entrapment rate and good controlled-release profile in vitro.	IVT	Preclinical rat models in vitro and in vitro.	[12]
Non-EAU, EIU	Succinated triamcinolone acetonide (TA-SA)	Supramolecular hydrogel with PECE nanoparticles.	Decreases neutrophil infiltration in anterior chamber. No noticeable side effects.	Topical	Preclinical rabbit models in vitro and in vitro.	[13]
Non-EAU, EIU	Triamcinolone acetonide	PLGA-chitosan nanoparticles.	Controlled drug release for 100 h. Excellent anti-inflammatory activity. Significantly reduced the secretion of IL-6.	Subconjunctival injection	Preclinical rabbit models in vitro and in vitro.	[14]
EAU	Rapamycin	EVs derived from mesenchymal stem cells (MSC-sEVs).	Excellent biocompatibility. Limited drug release kinetics and toxicity data.	Subconjunctival injection	Preclinical mouse models in vitro and in vivo.	[15]
Non-EAU, EIU	Tacrolimus	PNV	Prevented breakdown of the blood aqueous barrier. Sustained release of Tacrolimus for 12 h.	Topical	Preclinical rabbit models in vitro and in vivo.	[16]
Non-EAU, EIU, Anterior uveitis	Beclomethasone Dipropionate	Cubosomes and Cubosomal Gels	Significant controlled release. Enhanced corneal permeability. Greater relative bioavailability. Great encapsulation efficiency. High ocular tolerability.	Topical	Preclinical rabbit models in vitro and in vivo.	[17]
EAU	Copper–zinc superoxide dismutase (SOD1)	Multilayer polyion complex nanoparticles of SOD1	Penetrates interior eye structures more effectively than SOD itself. Retains enzyme activity in the eye longer. Restores antioxidant activity in the eye.	Topical	Preclinical rabbit models in vitro and in vivo.	[18]
EAU	Dexamethasone	Dexamethasone sodium phosphate supramolecular hydrogel composed of Dex and calcium ion	Well tolerated without complications of fundus blood vessel tortuosity or lens opacity. Downregulation of Th1 and Th17.	IVT injection	Preclinical rat models in vitro and in vivo.	[19]
Non-EAU, Carrageenan-induced	Triamcinolone acetonide	PEG-*b*-PCL and PEG-*b*-PLA micelles	High drug loading and drug encapsulation efficiencies. Hydrogel slowed the drug release rate (42% of drugs released in one week compared vs. ∼95%).	Topical	Preclinical rabbit models in vitro and in vivo.	[20]
EAU	Prednisolone	SMEDDS	Stable and sustained drug release. No irritation.	Topical	Preclinical rabbit models in vitro and in vivo	[21]
Non-EAU, EIU, Anterior uveitis	Ibuprofen	Hydrogel	Good cytocompatibility. Excellent ocular biocompatibility when instilled topically. Therapeutic efficacy comparable to current treatment.	Topical	Preclinical rabbit models in vitro and in vivo	[22]
EAU	Sirolimus (SRL)	Implant	Safe. Reduced inflammation.	Topical	Preclinical rabbit model in vivo	[23]
Non-EAU, EIU	Cyclosporin	In situ gel (PolyGel™) PCBCL-b-PEG-b-PCBCL	CyA’s t1/2 is 87% longer compared to Restasis^®^. Showed comparable profile to Restasis.	Topical	Preclinical rabbit model in vivo	[24]
Non-EAU, Anterior uveitis	Prednisolone phosphate and triamcinolone acetonide phosphate	PEG-liposomal formulation	Effective and sustained anti-inflammatory action (superior to eye drops). Little risk of globe injury compared to peribulbar injections. No intraocular penetration. No risk of endophthalmitis. Not powered to study adverse effects.	Subconjunctival injection	Preclinical rabbit model in vivo	[25]
Non-EAU, Anterior uveitis	Triamcinolone acetonide	Cationic nanostructured lipid carriers	Enhanced ocular bioavailability. No cytotoxicity and non-irritant. Could be retained inside cells for 24 h. Efficacious in uveitis treatment at a much lower concentration (0.1%) of drug.	Topical	Preclinical goat models ex vivo and in vitro	[26]
Non-EAU, EIU, Anterior uveitis	Dexamethasone	PCL-PEG-PCL micelles	More trials are needed. More prolonged follow-ups are needed. At 24 and 36 h PCL-PEG-PCL displayed a better inhibitory effect than market eye drops, but not significantly.	Topical	Preclinical rabbit models in vitro and in vivo	[27]
Non-EAU, EIU, Anterior uveitis	Flurbiprofen	Polypseudorotaxane hydrogels with Soluplus micelles	Excellent biocompatibility and ocular tolerance. Significantly suppressed intraocular inflammation at reduced administration frequency.	Topical	Preclinical rabbit models in vitro and in vivo	[28]
Non-EAU, EIU, Anterior uveitis	Dexamethasone	Nanoparticle	Good ocular tolerance. No changes in corneal thickness or intraocular pressure. Low cytotoxicity in human corneal epithelial cells at drug concentrations up to 1 mM after 24 h. Reduced cell viability after 48 h and 72 h.	Topical	Preclinical rabbit models in vitro and in vivo	[29]
Non-EAU, Posterior uveitis	Everolimus	Soluplus^®^: grafted copolymer of PVCL–PVA–PEG nanomicelles	High encapsulation efficiency. Sustained release of everolimus. Remained in the circulatory system for a longer duration. Significantly higher permeation across goat cornea. Improved drug bioavailability and accessibility.	Topical	Preclinical models ex vivo and in vitro	[31]
EAU, Posterior uveitis	Triamcinolone acetonide	Glycosylated triamcinolone acetonide hydrogelator hydrogel	Downregulated Th1 and Th17 responses. No cytotoxicity on ARPE-19 or RAW264.7 cells up to 600 uM. Toxic effects from TA suspension at 69 nmol/per eye.	IVT injection	Preclinical rat animal model in vivo	[30]
EAU, Posterior uveitis	Rapamycin	MET-RAP nanoparticle eyedrops	Activity like dexamethasone eye drops. Reduced RORγt and increased Foxp3 expression in IL-10.	Topical	Preclinical mouse and rabbit models in vitro and in vivo	[32]
Non-EAU, Endophthalmitis	Gatifloxacin	Nanosponge	Decreased ocular pathology and inflammation. Lack of toxicity. In vivo stability.	IVT injection	Preclinical murine and rabbit models in vitro and in vivo	[34]
Non-EAU, Endophthalmitis	Azithromycin or triamcinolone acetonide	Nanoparticle	Sustained release of drug for 300 h. Exhibited antimicrobial effects against Gram-positive and Gram-negative bacteria. Synergistic effects.	Topical	Preclinical mouse models in vitro and in vivo	[36]
Non-EAU, Postoperative uveitis	Curcumin	Double-headed polyester NPs with PLGA-GA_2_-CUR	Similar protection as with carprofen. Increased oral bioavailability of curcumin.	Oral	Preclinical adult male beagle model	[37]
Non-EAU, Postoperative endophthalmitis	Levofloxacin	Thermosensitive chitosan-based hydrogel	Displayed sustained-release profile. Long-term antibacterial property.	Cell culture assay	Preclinical rabbit epithelial cells in vitro	[38]
Non-EAU, Postoperative endophthalmitis	Predisolone acetate and levofloxacin	Chitosan-gelatin-based hydrogel containing NPs	Themosensitive and biocompatibility. Sustain drug-release properties.	Topical	Preclinical rabbit models in vitro and ex vivo	[39]
Non-EAU, Postoperative endophthalmitis	Bromfenac sodium (anti-inflammatory drug)	AuAgCu_2_O-BS NPs	Superior biocompatibility with low cytotoxicity. Metal ions could eliminate MDR bacteria (MRSA) effectively in vitro/vivo. Controlled the thermal damage to the surrounding ocular structure. Did not influence intraocular pressure; no significant toxicity.	Ocular injection	Preclinical rabbit model in vivo	[40]

AuAgCu_2_O-BS NPs: AuAgCu_2_O-bromfenac sodium nanoparticles. DSP-Zn-NP: Divalent zinc ion nanoparticle. EAU: Experimental autoimmune uveitis. EIU: Endotoxin induced uveitis. MET-RAP: Molecular Envelop Technology (N-palmitoyl-N-monomethyl-N,N-dimethyl-N,N,N-trimethyl-6-O-glycolchitosan)—Rapamycin. mPEGhexPLA: Methoxy-poly(ethylene-glycol)-hexyl substituted poly-lactic acid. mPEG-PLGA: methoxypoly(ethyleneglycol)-poly(dl-lactideco-glycolic acid). NP: nanoparticle. PCBCL-b-PEG-b-PCBCL: poly(α-carboxylate-co-α-benzylcarboxylate-ε-caprolactone)-block-poly(ethylene glycol)-block-poly(α-carboxylate-co-α-benzylcarboxylate-ε-caprolactone). PCL-PEG-PCL: polycaprolactone-polyethylene glycol-polycaprolactone. PECE: poly (ethylene glycol)-poly (ɛ-caprolactone)-poly (ethylene glycol). PEG-b-PCL: poly(ethylene glycol)-block-poly(ε-caprolactone). PEG-b-PLA: poly(ethylene glycol)-block-poly(lactic acid). PLGA: Poly(lactide-co-glycolide). PLGA-GA2-CUR: gambogic acid—coupled polylactide-co-glycolide—curcumin. PNV: Proglycosome nanovesicles. PVCL–PVA–PEG: polyvinyl caprolactam– polyvinyl alcohol–polyethylene glicol. SMEDDS: self-microemulsifying drug delivery systems.

**Table 4 pharmaceutics-15-01952-t004:** Biodegradable implants for uveitis.

Disease	Drug	Advantages and Considerations	Administration Route	Stage	Reference
Non-infectious uveitis	Dexamethasone (DEX)	Improved best-correct visual acuity, central retinal thickness, and vitreous haze. After 2< inserts, 75% of eyes underwent cataract surgery. Similar safety profile to other studies. Clinicians should consider an individualized plan for their specific needs.	Intravitreal implant (Ozurdex)	Clinical	[41]
Non-infectious uveitis	Dexamethasone	Similar outcomes with single DEX or multiple DEX injections. Does not affect systemic immunomodulatory therapy.	Intravitreal implant (Ozurdex)	Clinical	[42]
Non-infectious uveitis	Dexamethasone	None suffered from postoperative macular edema. No severe anterior chamber reactions. Postoperative visual acuity improved to varying degrees.	Intravitreal implant (Ozurdex)	Clinical trial	[43]
Posterior uveitis	Dexamethasone	Immediate improvement in vision-related functioning post implantation. Need for reimplantation every 6 months. Not permanent treatment due to adverse effects and disease progression (e.g., cataract progression). Small cohort size *n* = 3.	Intravitreal implant (Ozurdex)	Clinical trial	[44]
Posterior uveitis	Dexamethasone	Improved visual acuity and lower central retinal thickness maintained at 6 months. Did not worsen cataract development. Caused mild increase in intraocular pressure.	Intravitreal implant (Ozurdex)	Clinical trial	[45]
Posterior uveitis	Dexamethasone	Good safety profile. No systemic side effects. Ocular hypertension in 28% of subjects. Cataract surgery <1 year post-initial injection in 40% phakic subjects.	Intravitreal implant (Ozurdex)	Clinical	[46]
Infectious posterior uveitis	Dexamethasone	Effective in patients with contraindications for systemic corticosteroids.	Intravitreal implant (Ozurdex)	Clinical trial	[47]
Infectious posterior uveitis	Dexamethasone	Treat with appropriate anti-tuberculosis treatment and anti-inflammatory therapy. DEX-implant is safe and potent.	Intravitreal implant (Ozurdex)	Case-report	[48]
Infectious intermediate uveitis	Dexamethasone	Safe and efficacious. No recurrences or need for multiple injections.	Intravitreal implant (Ozurdex)	Clinical trial	[49]
Infectious intermediate and posterior uveitis	Dexamethasone	No worsening of the primary disease, emergence of a new choroiditis lesion, or retinal detachments. Improved visual acuity at 3 and 6 months. Risk of developing cataracts and glaucoma. Effective and relatively safe.	Intravitreal implant (Ozurdex)	Clinical trial	[50]
Intermediate and posterior uveitis	Dexamethasone	Increase intraocular pressure and cataract formation. Provides localized corticosteroid treatment. Adequate alternative to oral prednisone.	Intravitreal implant (Ozurdex)	Clinical trial	[51]
Noninfectious intermediate or posterior uveitis	Dexamethasone	Implant effectiveness comparable to systemic corticosteroid. Safe.	Intravitreal implant (Ozurdex)	Clinical trial	[52]
Intermediate and posterior uveitis	Dexamethasone	Cataracts and increased intraocular pressure side effects. No new safety concerns arise from the use of multiple implants.	Intravitreal implant (Ozurdex)	Clinical trial	[53]
Posterior and panuveitis	Dexamethasone	No serious ocular or systemic side effects. Improved case-corrected visual acuity, reduced macular edema, and reduced central foveal thickness.	Intravitreal implant (Ozurdex)	Clinical trial	[54]
Postoperative uveitis	Dexamethasone	Less postoperative inflammation and cystoid macular edema. Minimal adverse effects. Small sample size. Non-documentation of preoperative CME.	Intravitreal implant (Ozurdex)	Clinical trial	[55]
Postoperative refractory panuveitis	Dexamethasone	Significant improvement in best-corrected visual acuity. No improvement in central retinal thickness or grade of anterior chamber cells. Only *n* = 7.	Intravitreal implant (Ozurdex)	Clinical trial	[56]

**Table 5 pharmaceutics-15-01952-t005:** Biodegradable nano-based DDS for neuro-ophthalmologic diseases and retinal ganglion cell death.

Disease	Drug	DDS	Advantages and Considerations	Administration Route	Stage	Reference
TON	Ciliary neurotrophic factor + FK506 (immunosuppressant)	Thermosensitive hydrogel + mpolymeric micelle (for FK506)	Sustained release. RGC protection. Unclear administration route. Unclear impact on vision (transparency).	Not administered—smeared at the injury site	Preclinical (rabbit)	[60]
	Deferoxamine (DFO)	HA based NP	Useful for the delivery of drugs (like DFO) with short half-lives and poor bioavailability in vivo. Reduced cytotoxicity.	IVT	Preclinical (rat)	[61]
	PACAP38	Exosome	Localized and controlled drug delivery with anchor peptides. Promoting axon and nerve regeneration. Exosomes have low immunogenicity and can bypass the endosomal-lysosomal pathway due to their bilayer. Smaller exosomes more easily reach the retina.	IVT	Preclinical (rat)	[62]
	Methylene Blue (MB)	Hydrogel	Targeting various aspects of the same disease.	Small-gauge needles	Preclinical (in vitro human retinal pigment epithelial cells)	[63]
Unspecified optic neuropathy	Polydopamine (ROS scavenging) + Brimonidine	NP	Multiple effects on the mitochondrial, oxidative, and microglial environments of RGCs lead to overall RGC protection. Allows for synergistic delivery of drugs to targets of interest.	Unclear	Preclinical (optic nerve crush model with mice)	[64]
Unspecified optic neuropathy	NGF + BDNF	Magnetic NPs	MNPs allow self-localization of drugs to the retina and protection from rapid degradation of NT factors. Can re-assess many drugs that show promise in neuroprotection or ocular therapy but did not succeed in trials using a DDS approach.	IVT	Preclinical (cell lines, zebrafish larvae)	[65]
RGC death secondary to retinal degeneration	Melatonin	Ethylcellulose NPs	Slower release and greater penetration.Non-invasive route of administration to the posterior segment.	Topically	Preclinical (rabbits)	[66]
Glaucoma and partial optic nerve transection	Curcumin	Succinate NPs	Useful for drugs with poor solubility and low bioavailability. Protective against hypoxia and glutamate-induced toxicity.	Topical	Preclinical (cell lines, rats)	[67]
ONC	siRNA targeting caspase-3	polybutylcyanoacrylate NPs	Non-viral gene therapy. Selective silencing of target genes.	Intraocular	Preclinical (rats)	[68]
RGC death secondary to retinal degeneration	siRNA	Positively charged lipid NPs	Targeting specific layers (i.e., RGC layer). Possibility to optimize the NP charge of specific drug delivery.	IVT	Preclinical (cell lines)	[69]
ONC	Trichostatin A (a histone deacetylase inhibitor)	Polyethylene glycolylated liposomes	Liposomes reached the inner retina and were detected up to 10 days post-injection, with a peak at 3 days. Significant decrease in reactive gliosis.	IVT	Preclinical (mice)	[59]
LHON	Idebenone	PLGA Microspheres	Size adjustable for ease of administration. Site-specific, maintained release. High encapsulation capacity. No cytotoxicity.	IVT	Preclinical (in vitro)	[70]
Unspecified optic neuropathy and RGC death secondary to retinal damage	Multiple mTOR pathway stimulating biologics (CNTF, IGF-1, and LNOM)	Liposome	Combining multiple biologics with controlled release. Protection from degradation and internal clearance mechanisms. Improved electrophysiological outcomes when combined with transplanted RGCs.	IVT	Preclinical (mouse model of NMDA) induced RGC death	[71]
ONC and RGC deaths secondary to retinal degeneration	CNTF + oncostatin N (NT factors)	NP	Sustained release. RGC protection and vision preservation. Combined treatments of neuroprotective agents with antioxidants or anti-inflammatory.	IVT	Preclinical (rats)	[65]

ATP: adenosine triphosphate. BDNF: brain derived neurotrophic factor. CNTF: ciliary neurotrophic factor. DDS: drug delivery system. DFO: deferoxamine. FK506: Tacrolimus. HA: hyaluronic acid. IGF-1: insulin-like growth factor 1. IV: intravenous. IVT: intravitreal. LHON: Leber’s hereditary optic neuropathy. LNOM: lipopeptide N-fragment osteopontin mimic. MB: Methylene Blue. MNP: magnetic nanoparticle. mTOR: mammalian target of rapamycin. NGF: nerve growth factor. NMDA: N-methyl-D-aspartate. NP: nanoparticle. NT: neurotrophic factor. ON: optic nerve. ONC: optic nerve crush. PACAP38: Pituitary adenylate cyclase-activating polypeptide 38. PLGA: poly(lactic-co-glycolide). RGC: retinal ganglion cells. ROS: reactive oxygen species. siRNA: small interfering ribonucleic acid. TON: traumatic optic neuropathy.

## Data Availability

Not applicable.

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
