# Peer review of "A New Era in Ocular Therapeutics: Advanced Drug Delivery Systems for Uveitis and Neuro-Ophthalmologic Conditions"

_pharmaceutics, 2023, doi:10.3390/pharmaceutics15071952_

Round 1

Reviewer 1 Report

Nice Job. The only critique I dare to provide is that the review is too long.

Maybe the author could reduce the text of the central part of the work when they generally discuss DDS applied for eye disease

Author Response

Dear Reviewer,

Thank you for your valuable comments and constructive feedback on our manuscript. We appreciate the time and effort you have dedicated to providing us with a detailed assessment, which will undoubtedly help improve the quality of our work.

We sincerely thank you for recognizing our work. Concerning the length of the review, we understand your perspective and have since revised the manuscript, aiming to present the information more concisely without losing crucial details. This change should improve readability and maintain the comprehensive nature of the review.

Once again, we would like to express our gratitude for your insightful feedback. We believe that the revised manuscript is considerably improved and hope that it meets your approval.

Reviewer 2 Report

The manuscript “A New Era in Ocular Therapeutics: Biodegradable Nano-Based DDS for Uveitis and Neuro-Ophthalmologic Conditions” seems interesting and informative. The authors' comprehensive review of preclinical and clinical studies from 2017 to 2023 demonstrates their dedication and effort. However, I kindly suggest considering the following points for further improvement before publication:

1) It will be better to define the abbreviation in the title (DDS).

2) Font size consistency: It would greatly enhance the readability and comprehension of the manuscript if the authors ensure uniformity and consistency of the font size throughout. Specifically, I recommend reviewing page 28, lines 858-865, and page 37, lines 1069-1074, as the font size appears to be too small. Addressing this matter consistently across the manuscript will significantly benefit readers.

3) Discussion of limitations/challenges: To provide a comprehensive overview of the field, it is important to discuss the limitations and challenges associated with existing strategies for ocular therapeutics development. Including a dedicated section on this topic will enhance the manuscript by highlighting areas that require further improvement and considering potential drawbacks.

4) Inclusion of future perspectives and research directions: To augment the manuscript's value, I suggest incorporating a separate section that discusses future perspectives and research directions. This addition will offer insights into potential advancements and upcoming research areas in the field of ocular therapeutics, providing valuable guidance to researchers and practitioners.

Moderate editing of English language required.

Author Response

Thank you for your valuable comments and constructive feedback on our manuscript. We appreciate the time and effort you have dedicated to providing us with a detailed assessment, which will undoubtedly help improve the quality of our work.

We acknowledge your suggestion regarding the abbreviation in the title. We have revised the title to: "A New Era in Ocular Therapeutics: Advanced Drug Delivery Systems for Uveitis and Neuro-Ophthalmologic Conditions."

We appreciate your detailed observation regarding font consistency. We have thoroughly reviewed the manuscript and corrected the font size of the text associated with the two tables, ensuring a uniform and consistent appearance throughout the manuscript.

We agree with your suggestion about discussing the limitations and challenges associated with ocular therapeutics development. We have now included a section (section 7 - Challenges, future perspectives, and future research directions) devoted to this topic, which we believe will add depth to the manuscript.

Your recommendation to include a section on future perspectives and research directions is excellent. We have added a new section (section 7 - Challenges, future perspectives, and future research directions) addressing this, providing readers with insight into potential advancements and upcoming research areas in the field of ocular therapeutics.

Once again, we would like to express our gratitude for your insightful feedback. We believe that the revised manuscript is considerably improved and hope that it meets your approval.

Reviewer 3 Report

This review paper focuses on biodegradable DDS for ocular drug delivery, specifically to uveitis and neuro-ophthalmological diseases.  Overall, it is comprehensive. There are several comments to be addressed before publication.

- the title is not appropriate. Authors provide many DDS which are not nano-sized, such as Ozurdex.

-  the novelty should be distinguished from recent published paper:  Clinical Translation of Prophylactic Drug Delivery Systems for Posterior Capsule Opacification. Pharmaceutics, 2023, 15, 1235.

- In Figure 2, the anatomical barriers were not clearly illustrated in the eyes. It is better to locate those barriers in the right location in the eye.

- For treating endophthalmitis, multiple drugs (e.g., eyedrops) are used. the recent paper Depot unilamellar liposomes to sustain transscleral drug co-delivery for ophthalmic infection therapy. Journal of Drug Delivery Science and Technology. 2023. 86. 104629, is one of good example of illustrating the strategy of surmouting barrier hurdles.

No.

Author Response

Dear Reviewer,

Thank you for your valuable comments and constructive feedback on our manuscript. We appreciate the time and effort you have dedicated to providing us with a detailed assessment, which will undoubtedly help improve the quality of our work.

You raise a valid point about the title. In light of your comment, we have revised the title to more accurately reflect the content of our paper: “A New Era in Ocular Therapeutics: Advanced Drug Delivery Systems for Uveitis and Neuro-Ophthalmologic Conditions”.

Thank you for pointing out the comparison to the recently published paper titled "Clinical Translation of Prophylactic Drug Delivery Systems for Posterior Capsule Opacification." We understand your concern regarding the novelty of our manuscript compared to the recently published work you mentioned. While this article is indeed an interesting piece of work, we'd like to clarify the distinct scope of our manuscript.

The aforementioned paper focuses exclusively on the use of DDS in preventing Posterior Capsule Opacification (PCO), which is a single, specific ocular condition. Our manuscript, however, offers a comprehensive review of the use of biodegradable DDS in treating a broader range of ocular diseases including uveitic and neuro-ophthalmologic disease.

Furthermore, the diseases we address in our article, such as endophthalmitis and optic neuropathy, are often irreversible and severe, requiring urgent and highly effective treatments. On the other hand, and from our perspectives as ophthalmic surgeons, PCO, though a common sequelae after cataract surgeries, is relatively less severe and can be effectively managed with treatments like YAG laser capsulotomy.

Therefore, while the paper you referenced provides valuable insights into a specific application of DDS, we believe our work offers a broader perspective on the potential of biodegradable DDS in managing multiple, often more severe and irreversible, ocular conditions. We hope this clarifies the novelty and comprehensiveness of our manuscript in comparison to the mentioned study.

We also agree that Figure 2 could provide a clearer illustration of anatomical barriers in the eyes. This figure has been revised to better locate these barriers.

Your suggestion to incorporate the recent paper on depot unilamellar liposomes is appreciated. We have updated our manuscript to include a discussion on this strategy for overcoming barrier hurdles in treating endophthalmitis.

Once again, we would like to express our gratitude for your insightful feedback. We believe that the revised manuscript is considerably improved and hope that it meets your approval.

Round 2

Reviewer 2 Report

The authors have revised the manuscript as per my comments and suggestions, thus I recommend it for publication.

Moderate editing of the English language is required.